# Gender-Differentiated Analysis of the Correlation between Active Commuting to School vs. Active Commuting to Extracurricular Physical Activity Practice during Adolescence

**DOI:** 10.3390/ijerph18115520

**Published:** 2021-05-21

**Authors:** Nuria Castro-Lemus, Cristina Romero-Blanco, Virginia García-Coll, Susana Aznar

**Affiliations:** 1FENIX Research Group, Faculty of Sports Sciences, University of Sevilla, 41013 Sevilla, Spain; 2PAFS (Physical Activity and Health Promotion) Research Group, Faculty of Nursing, University of Castilla-La Mancha, 13071 Ciudad Real, Spain; cristina.romero@uclm.es; 3PAFS Research Group, Faculty of Sports Sciences, University of Castilla-La Mancha, 45071 Toledo, Spain; Virginia.garcia@uclm.es

**Keywords:** active commuting, extracurricular physical activity, gender

## Abstract

Active commuting to school in children and adolescents can help achieve compliance with the World Health Organization (WHO) recommendations for physical activity. This study aimed to evaluate the relationship between the mode of transport used to go to school and the mode of transport used to go to extracurricular sports practice. Multistage random cluster sampling was conducted to include 128 schools with the participation of 11,017 students between the ages of 5 and 19. Participants completed the survey of sports habits designed by the National Sports Council. The results revealed that the mode of transport used to go to school is significantly related to the mode of transport used to go to sports practice. A total of 54.3% of students aged 5 to 19 years walk to school. A total of 23.7% of boys walk and 7.9% bike to extracurricular physical activities vs. 24.1% of girls who walk. The fact that girls only walk to extracurricular physical activities implies that the organized sports activities were nearby. Therefore, it seems crucial to have a wide range of physical activities on offer locally to promote extracurricular physical activity participation for girls.

## 1. Introduction

Physical activity is associated with reduced risk of obesity, lower blood pressure, and improved mental wellbeing among children [1]. Sedentary time can be a risk factor for non-communicable diseases, and to date, it is not clear if this effect is independent of physical activity in children [2,3,4]. Therefore, physical activity guidelines state that all children and young people should engage in at least an hour per day of moderate-to-vigorous-intensity physical activity (MVPA) and limit sedentary time [5]; however, considerable proportions of children do not meet these guidelines [6].

Nevertheless, during adolescence, daily physical activity decreases [7], and this decrease is more pronounced among females [8]. More than 80% of school-aged adolescents worldwide—specifically, 85% of females and 78% of males—do not achieve the minimum level recommended of one hour of physical activity daily [9]. Hence, the promotion of physical exercise among children and youth has been converted into a public health priority [10].

Physical activity is considered as all body movement that expends energy [11], and therefore, activities such as riding a bike, dancing, or active travel are calculated to determine the level of physical activity. More and more scientific evidence exists that demonstrates the benefits of incorporating active travel into adolescents’ daily lives. Therefore, adolescents are encouraged to choose active modes of transport that increase their daily physical activity time.

Ecological models of behavior change indicate that behavior change depends on the interaction between people and the environment in which they live, learn, work, and play [12]. Currently, scientific evidence suggests that transport is one of the potential sources of continuous moderate physical activity [13,14]. Active commuting guarantees mental health benefits [14], improved academic and cognitive performance [15], and decreases obesity, and thereby decreases the probability of developing cardiovascular diseases [16,17]. However, what is the connection among the transport used for these types of activities?

The Helena study carried out in 10 European cities estimated that adolescents spent about 30 min a day walking, with no gender differences. In terms of cycling, boys were more likely to use bicycles than girls [18]. The ANIBES study carried out in Spain showed that motorized transport was used by more than 30% of the child population, with gender differences; 37.3% of boys compared to 33.3% of girls. In the case of adolescents, around 71% walked to school [19].

Commuting to other locations has not been studied in depth. It seems that those students who walked or cycled to school were more likely to use the same type of commuting to other places [20]. Active commuting may change for a compulsory behavior (such as going to school) versus a chosen one (going to one’s chosen extracurricular sports activity). Moreover, extracurricular sports activity includes exercise, and adolescents may be more inclined to actively commute, or maybe not. Active transportation needs to be encouraged and all possibilities should be explored, in particular those who are structured within one’s weekly timetable (school and extracurricular sports activity).

This study aimed to evaluate the relationship between the mode of transport used to go to school and the mode of transport used to go to extracurricular sports practice, and whether this relationship differs by gender. Analyzing the relationship between these two modes of transportation for different behaviors will provide us with some useful information.

## 2. Materials and Methods

The present study is a quantitative, correlational, cross-sectional study on the mode of transport used by youth of the autonomous community of Castilla-la Mancha (Spain) which has five provinces. A multistage random cluster sampling adjusted by province was performed. Participating schools were recruited by sending letters detailing the objective of the study and their proposed inclusion. The study was carried out in public, private, and state-subsidized schools, with a total of 128 participating schools distributed as follows: Toledo (2413 students), Ciudad Real (2954 students), Albacete (2696 students), Cuenca (1598 students), and Guadalajara (1356 students).

The total number of participants was *n* = 11,017, with a mean age of 12.56 (±2.58). A total of 50.1% of students were in O-level and 49.9% in A-level. The total population aged 5 to 19 years in this autonomous community is 320,267 [21], and, therefore, our results have a 99% confidence level and a margin of error of ±1.21%. The distribution by sex for our sample was 49.6% males and 50.4% females (Table 1).

### 2.1. Questionnaire

An ad hoc questionnaire was used to quantify the physical activity habits of school-aged youth in Castilla-la Mancha. This questionnaire was based on the “Study of sports habits, promoted by the High Sports Council of Castilla-la Mancha”. For this research, the sociodemographic item “gender” was used along with items on the mode of transport used to go to school and the mode of transport used to go to extracurricular sports activities. The possible answers to both questions were categorical with the following options, “on foot”, “bike, skateboard, scooter, or skates”, “public transport”, or “car or motorbike”.

### 2.2. Statistical Analysis

The statistical package SPSS 25.0 was used for all analyses. A descriptive analysis of the variables was conducted, with absolute and relative frequencies as measures of distribution. Analyses were also conducted separately by gender. A bivariate analysis was conducted for both variables and their relationship with gender using Pearson’s chi-square test. To assess the degree of association between the variables “transport used to go to school” and “transport used to go to sport”, a contingency analysis was carried out, estimating only the degree of significance lower than 0.01 and a 99% confidence level between both variables. The level of significance was set at *p* < 0.05.

## 3. Results

The initial hypothesis was that the mode of transport used to go to school was correlated with the mode of transport to go to sport. To test this hypothesis, we analyzed the distribution of frequencies by gender. As shown in Table 2, this distribution is different for both variables, assuring us of the suitability to carry out a contingency analysis, and separating the participants by gender.

The chi-square test results clearly show the existence of significant differences between males and females corresponding to a degree of significance of *p* < 0.001 (Table 2). Therefore, we can assert with a 99% confidence level that there is a significant difference between males and females.

Reading the same table (Table 2) in terms of the mode of transport used to go to school, most students do so on foot (first), followed by car or motorcycle (second). Females use car or motorbike the most as a mode of transport (16.2% of males compared with 17.4% of females). In terms of transport used to go to sports, 23% of males responded they go on foot and 22.5% by car or motorbike. In contrast, 24% of females responded they go on foot, and 17.7% use a car. Moreover, 97.1% of students took less than 30 min to commute to school from home, and 96.6% of students took less than 30 min to commute to their extracurricular physical activity practice.

Next, we analyzed the existence of a correlation between “mode of transport used to go to school” and “mode of transport used to go to sport”, and, considering the above, it made sense to differentiate by gender.

In the case of females, using the contingency coefficient (Table 3), we can observe that *p* < 0.001, so we can say with a 99% confidence level that a correlation exists between both variables. That is, a significant tendency exists, albeit low (0.299), that females use the same mode of transport to go to school as to go to sports in all cases, except in the case of public transport to school. In this case, those girls that go to school via public transport go on foot to sports (Table 3).

In terms of males, the same correlation (*p* < 0.001) exists; however, in this case, it is moderate (0.374) and without exceptions. Males use the same mode of transport to go to school as to go to sport (Table 4).

## 4. Discussion

The present study results show that the mode of transport used to go to school is related to the mode of transport used to go to sports practice. In addition, we observed that the majority of students aged 5 to 19 years walk to school, although, by gender, females use motorized transport more. To our knowledge, this is the first study to analyze active transport in our community in a representative sample of the population between 5 and 19 years of age for both travel to school and to extracurricular sporting activities. Moreover, there are no studies that have compared both types of travel and their differences by gender. In fact, there are few studies that have analyzed commuting to sporting activities.

This study aimed to demonstrate and analyze the relationship between the mode of transport to school and the mode of transport to sport and their relationship with gender. Our study contributes to examining and giving a possible explanation for the lower rate of practicing sport among females compared to males [22]. Considering that most females go to sports on foot, and those who take public transport to school also go to sports on foot, we are led to believe that females’ sport is probably that which is available in their close surroundings. This consideration is supported by the social–ecological model, which explains how the practice of a physical activity is conditioned by biological, environmental, and social factors [20,23].

Active commuting was used by 55.8 of the students analyzed, mostly on foot. Gálvez-Fernández et al., after reviewing the mode of travel in 34 analyses conducted over 7 years in Spanish students, found that 60% of the participants actively commuted to school. In this study, active commuting was slightly lower than in the review in terms of transport to school; however, it decreased even more in the case of travel to extracurricular sports activities, being less than 50% [24].

Regarding travel to extracurricular activities, Drake et al. examined the relationship between commuting and obesity values and found that those students who walked or cycled to school were more likely to use the same type of commuting to other places. However, in that study, male gender was identified as a positive predictor of obesity while sports participation was a negative predictor [20]. In our case, we did not analyze physical activity and obesity in our sample; girls were more likely to use motorized commuting, and this could be a predictor of obesity. More factors should be investigated, as proposed by the ecological model.

In general, we can assert, similar to other studies, that although in general terms the data appears to be similar, differences exist between males and females [25]. We can say that significant differences exist between females and males with respect to transport used to go to sport; males go almost without difference by foot or car and females go mostly on foot; although, in terms of going to sports, females take the skateboard or the bicycle more frequently than when going to school, being less frequent than the males. Overall, among schoolchildren, whether male or female, the most used mode of transport to go to school is walking, an active mode of commuting. Similar findings have also been observed in several other studies [26], for example, a study by Villa-González, Ruiz, and Chillón in which a total of 57.2% of schoolchildren made the journey on foot, 40.3% used a car, 1.2% used a motorcycle, 0.7% used the bus, and 0.6% used a bicycle [27].

Other studies, such as that by Rodríguez-Rodríguez, Cristi-Montero, Celis-Morales, Escobar-Gómez and Chillón [28], have also demonstrated that the principal mode of transport was by car for children (to school 64.9%) and adolescents (to school 50.2%). Only 11.0% and 24.8% of children and adolescents, respectively, walked to school. There is a need to increase children’s and adolescents’ physical activity levels. After-school physical activity (i.e., participating in organized physical activity at school and in the community) is a great opportunity to meet physical activity guidelines because it is associated with more physical activity and reduced sedentary time among both boys and girls [29,30].

Recent marked increases in the prevalence of obesity in Spanish children and youth have prompted public health authorities to emphasize the importance of achieving physical activity guidelines [31,32]. Public health interventions in this regard appear to be particularly needed for adolescent girls, among whom the prevalence of overweight and obesity (≥85th percentile body mass index for age) has risen to 34.9% (20.7% overweight and 14.2% obese) [33].

One study reported that the availability of physical activity resources within walking distance (within a 0.75 mile street-network buffer around adolescent girls’ homes) was positively associated with their physical activity levels [34]. These findings are in line with our results, as girls used walking as the main mode of transport to go to after school physical activities, which implies that the organized sports activities were nearby. Therefore, it seems crucial to have a wide range of physical activities on offer locally and, most importantly, to make sure this offer is promoted and the information about it spread widely.

### Limitations and Future Research

The type of study conducted allows associations of results to be estimated without being able to establish causal relationships. Factors that could have been taken into consideration, such as the distance between school and home, owning a bicycle, being accompanied to school by siblings or peers, road safety, etc., were not analyzed in this study. Future research should include these variables to make a more accurate approximation of the environmental factors related to active commuting. A longitudinal design would be appropriate to establish this. On the other hand, it would have been interesting to have further examined whether practicing sports among females is conditioned by the availability in their close vicinity to home.

## 5. Conclusions

In conclusion, the majority of young people between 5 and 19 years go to school on foot, but when by car or motorbike, females use this mode of transport more often. The mode of transport to go to sports activities for males is on foot or by bike indiscriminately, whereas females mostly do so on foot. The mode of transport used to go to school is related to the mode of transport used to go to sports, except for females who use public transport to go to school who also go to sports on foot. Finally, the sports activities in which females participate are those that are available nearby, that is, those which they can go to by walking. Therefore, the availability of local physical activities and their promotion to raise their visibility appear to be necessary strategies to encourage participation of adolescent females.

## Figures and Tables

**Table 1 ijerph-18-05520-t001:** Distribution of frequencies and percentages by gender.

Heading	Gender	Frequency	Percent	Valid Percentage	Cumulative Percentage
Valid	Male	5402	49.0	49.6	49.6
	Females	5486	49.8	50.4	100.0
	Total	10,888	98.8	100.0	
Missing	System	129	1.2		
Total		11,017	100.0		

**Table 2 ijerph-18-05520-t002:** Distribution of frequencies by sex and chi-square test.

Commuting	Gender	On Foot	Bike, Rollerblades, or Roller Skates	Public Transport	Car or Motorbike	Total	Pearson’s Chi-Square		Asymptotic Significance (Bilateral)
							Value	df	
School Transport	Males	27.1%	1.1%	5.4%	16.2%	49.8%	31.245 ^a^	3	0.000 *
	Females	27.2%	0.5%	5.1%	17.4%	50.2%			
	Total	54.3%	1.5%	10.6%	33.6%	100.0%			
Transport to sport	Males	23.7%	7.9%	1.2%	22.5%	55.3%	177.236 ^b^	3	0.000 *
	Females	24.1%	2.3%	0.7%	17.7%	44.7%			
	Total	47.8%	10.2%	1.8%	40.2%	100.0%			

^a^ 0 cells (0.0%) have an expected count less than 5. The minimum expected count is 79.25. ^b^ 0 cells (0.0%) have an expected count less than 5. The minimum expected count is 17.93. df: degrees of freedom * *p* < 0.001.

**Table 3 ijerph-18-05520-t003:** Contingency between the variables “mode of transport to school” and “mode of transport to sport” in females.

Mode of Transport to School			Transport to Sport					Contingency Coefficient	
			On Foot	Bike, Skateboard, Scooter, or Skates.	Public Transport	Car or Motorbike	Total	Value	Approx. Significance
Transport to school	On foot	C	947 **	71	23 *	442	1483	0.299	0.000 ***
		EC	799.6 **	74.4	21.9 *	587.1	1483.0		
	Bike, skateboard, scooter, or skates	C	7	12 **	1	9	29		
		EC	15.6	1.5 **	0.4	11.5	29.0		
	Public transport	C	131 **	8	9 *	70	218		
		EC	117.5 **	10.9	3.2 *	86.3	218.0		
	Car or motorbike	C	409	48	8	576 **	1041		
		EC	561.3	52.2	15.4	412.1 **	1041.0		
Total		C	1494	139	41	1097	2771		
		EC	1494.0	139.0	41.0	1097.0	2771.0		

C: Count. EC: Expected count. * A tendency exists among variables: the actual count is greater than the expected count. ** A greater tendency exists among variables. *** *p* < 0.001.

**Table 4 ijerph-18-05520-t004:** Contingency between the variables “mode of transport to school” and “mode of transport to sport” in males.

Mode of Transport to School			Transport to Sport					Contingency Coefficient	
			On Foot	Bike, Skateboard, Scooter, or Skates	Public Transport	Car or Motorbike	Total	Value	Approx. Significance
Transport to school	On foot	C	1032 **	252	30	554	1868	0.374	0.000 ***
		EC	797.9 **	269.9	38.3	761.8	1868.0		
	Bike, skateboard, scooter, or skates.	C	12	50 **	0	14	76		
		EC	32.5	11.0 **	1.6	31.0	76.0		
	Public transport	C	122	52 *	26 **	111	311		
		EC	132.8	44.9 *	6.4 **	126.8	311.0		
	Car or motorbike	C	312	146	15	732 **	1205		
		EC	514.7	174.1	24.7	491.4 **	1205.0		
Total		C	1478	500	71	1411	3460 *		
		EC	1478.0	500.0	71.0	1411.0	3460.0 *		

C: Count. EC: Expected count. * A tendency exists among variables: the actual count is greater than the expected count. ** A greater tendency exists among variables. *** *p* < 0.001.

## Data Availability

The data supporting reported results can be found in https://grupopafs.com/contacto/ (accessed on 24 March 2021).

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
