# Peer review of "Gender-Differentiated Analysis of the Correlation between Active Commuting to School vs. Active Commuting to Extracurricular Physical Activity Practice during Adolescence"

_ijerph, 2021, doi:10.3390/ijerph18115520_

Round 1

Reviewer 1 Report

The topic of active commuting is indeed an important research topic. The sample size is large in this study (although whether the sample is representative is unknown). However, there several are critical flaws in the paper.

  • The introduction provides a nice overview of physical activity benefits associated with active commuting, but fails to review the literature associated with the content of the article - gender related differences or comparisons between active school commuting and active commuting to after school activities.
  • The method section fails to fully describe the research design and sampling strategy/recruitment is unknown.
  • The analyses are descriptive only. There does not appear to be any consideration for the many other factors (e.g. age, physical environment, distance from home, bike ownership, etc.) that contribute to travel behavior (despite being framed around an ecological model).
  • The results do not substantially contribute to the body of knowledge. There is already an understanding that children who identify as males generally use more active forms of commuting compared to those who identify as females. Also, the difference between active commuting in the morning and to after-school activities appears to be similar to other published research.

Author Response

The topic of active commuting is indeed an important research topic. The sample size is large in this study (although whether the sample is representative is unknown). However, there several are critical flaws in the paper.

AUTHORS: We welcome your comments. Thank you for your time and effort. They allow us to improve our manuscript.

REVIEWER: The introduction provides a nice overview of physical activity benefits associated with active commuting but fails to review the literature associated with the content of the article - gender related differences or comparisons between active school commuting and active commuting to after school activities.

AUTHORS: Thank you for your comments. More information on gender differences in active commuting has been included in the introduction section. The studies included are the following:

“The Helena study, carried out in 10 European cities, estimated that adolescents spent about 30 minutes a day walking, with no gender differences. In terms of cycling, boys were more likely to use bicycles than girls.

Chillón P, Ortega FB, Ruiz JR, De Bourdeaudhuij I, Martínez-Gómez D, Vicente-Rodriguez G, Widhalm K, Molnar D, Gottrand F, González-Gross M, Ward DS, Moreno LA, Castillo MJ, Sjöström M; HELENA study group. Active commuting and physical activity in adolescents from Europe: results from the HELENA study. Pediatr Exerc Sci. 2011 May;23(2):207-17. doi: 10.1123/pes.23.2.207. PMID: 21633133.)

The ANIBES study carried out in Spain, showed that motorized transport was used by more than 30% of the child population, with gender differences: 37.3% of boys compared to 33.3% of girls. In the case of adolescents, around 71% walked to school.

Aparicio-Ugarriza R, Mielgo-Ayuso J, Ruiz E, Ávila JM, Aranceta-Bartrina J, Gil Á, Ortega RM, Serra-Majem L, Varela-Moreiras G, González-Gross M. Active Commuting, Physical Activity, and Sedentary Behaviors in Children and Adolescents from Spain: Findings from the ANIBES Study. Int J Environ Res Public Health. 2020 Jan 20;17(2):668. doi: 10.3390/ijerph17020668. PMID: 31968634; PMCID: PMC7014153.

Commuting to other locations has not been studied in depth. It seems that those students who walked or cycled to school were more likely to use the same type of commuting to other places.” Drake KM, Beach ML, Longacre MR, Mackenzie T, Titus LJ, Rundle AG, Dalton MA. Influence of sports, physical education, and active commuting to school on adolescent weight status. Pediatrics. 2012 Aug;130(2):e296-304. doi: 10.1542/peds.2011-2898. Epub 2012 Jul 16. PMID: 22802608; PMCID: PMC3408684. One recent study by Aibar Solana et al. reported that mothers-specific correlates of ACS included a positive association with children's extra-curricular activities organization. However, this fact was reported by mothers and not by adolescents. This is one of the reasons why we decided to compare these two behaviors.

Aibar-Solana, A.; Mandic, S.; Generelo, E.; Gallardo, L. O.; Zaragoza Casterad, J. Parental barriers to active commuting to school in children: does parental gender matter? Journal of Transport & Health. 2018, 9: 141-149. Doi: https://doi.org/10.1016/j.jth.2018.03.005.

REVIEWER: The method section fails to fully describe the research design and sampling strategy/recruitment is unknown.

AUTHORS: Thank you for your comments. The research design and sample recruitment have been included in the methodology section. The text included is:

“A multistage random cluster sampling adjusted by province was performed. Participating schools were recruited by sending letters detailing the object of the study and their proposed inclusion.

The study was carried out in public, private and state-subsidised schools, with a total of 128 participating schools distributed as follows: Toledo (2413 students), Ciudad Real (2954 students), Albacete (2696 students), Cuenca (1598 students) and Guadalajara (1356 students).”

REVIEWER: The analyses are descriptive only. There does not appear to be any consideration for the many other factors (e.g. age, physical environment, distance from home, bike ownership, etc.) that contribute to travel behavior (despite being framed around an ecological model).

AUTHORS: Thank you for your comments. You are totally right. We should have taken other factors into consideration. Therefore, the following paragraph has been included in the discussion section:

“Factors that could have been taken into consideration such as the distance between school and home, owning a bicycle, being accompanied to school by siblings or peers, road safety, etc., were not analyzed in this study.  Future research should include these variables to make a more accurate approximation of the environmental factors related to active commuting.”

REVIEWER: The results do not substantially contribute to the body of knowledge. There is already an understanding that children who identify as males generally use more active forms of commuting compared to those who identify as females. Also, the difference between active commuting in the morning and to after-school activities appears to be similar to other published research.

AUTHORS: Thank you for your comment. New paragraphs have been added in the discussion section to clarify our contribution in this manuscript. Although some studies have looked at gender (e.g. Rodríguez-Rodríguez et al (DOI:10.3390/ijerph17186864), Larsen et al. (DOI: 10.2105 / AJPH.2008.135319)our study has emphasized on the need to provide physical activity opportunities close by for girls. Mainly because we can see that they commute only walking to extra curricular physical activities and also that their participation level in them is lower than boys. Obviously, the possible causes for the different behavior of boys and girls need to be studied in depth. This will allow us to target this issue with a broader approach.

Reviewer 2 Report

The manuscript presents a very relevant theme for the health of children and adolescents. It presents important data that relate part of habits of active behavior for a given population with specific characteristics, and these results cannot be generalized.
The introduction is well founded, an adequate method to answer the proposed objective. I have suggestions for presenting the results, which are in the attached file
The manuscript requires minor corrections

Author Response

The manuscript presents a very relevant theme for the health of children and adolescents. It presents important data that relate part of habits of active behavior for a given population with specific characteristics, and these results cannot be generalized.

The introduction is well founded, an adequate method to answer the proposed objective. I have suggestions for presenting the results, which are in the attached file

The manuscript requires minor corrections

AUTHORS: Thank you for your comments. We will address each of them in the manuscript.

Reviewer 3 Report

The subject transport and school are very interesting and indeed a valid target when promoting physical activity in youth. Yet, this manuscript lacks in-depth analysis and substantiation of findings.

Although the numbers of participants are quite high, there is little information about influencing factors like age, primary/secondary school, distance between school and home, safety of crossings between home and school, having older brothers/sisters to guide younger children etc.

The introduction includes physical activity information related to household tasks, which of course are important, but not the most appropriate example when assessing PA in children.

Author Response

The subject transport and school are very interesting and indeed a valid target when promoting physical activity in youth. Yet, this manuscript lacks in-depth analysis and substantiation of findings.

AUTHORS: Thank you for your comment. New paragraphs have been added in the introduction and discussion sections to explore the analysis and findings in more detail.

“To our knowledge, this is the first study to analyze active transport in our community in a representative sample of the population between 5 and 19 years of age for both travel to school and to extracurricular sporting activities. Moreover, there are no studies that have compared both types of travel and their differences by gender. In fact, there are few studies that have analyzed commuting to sporting activities.

55.8 of the students analyzed used active commuting, mostly on foot. Gálvez-Fernández et al. after reviewing the mode of travel in 34 analyses conducted over 7 years in Spanish students found that 60% of the participants made active commuting to school. In this study, active commuting was slightly lower than in the review in terms of transport to school, however, it decreased even more in the case of travel to extracurricular sports activities, being less than 50%.”

Gálvez-Fernández P, Herrador-Colmenero M, Esteban-Cornejo I, Castro-Piñero J, Molina-García J, Queralt A, Aznar S, Abarca-Sos A, González-Cutre D, Vidal-Conti J, Fernández-Muñoz S, Vida J, Ruiz-Ariza A, Rodríguez-Rodríguez F, Moliner-Urdiales D, Villa-González E, Barranco-Ruiz Y, Huertas-Delgado FJ, Mandic S, Chillón P. Active commuting to school among 36,781 Spanish children and adolescents: A temporal trend study. Scand J Med Sci Sports. 2021 Apr;31(4):914-924. doi: 10.1111/sms.13917. Epub 2021 Jan 29. PMID: 33423302.

“Regarding travel to extracurricular activities, Drake et al. examined the relationship between commuting and obesity values and found that those students who walked or cycled to school were more likely to use the same type of commuting to other places. However, in that study male gender was identified as a positive predictor of obesity while sports participation was a negative predictor.”

Drake KM, Beach ML, Longacre MR, Mackenzie T, Titus LJ, Rundle AG, Dalton MA. Influence of sports, physical education, and active commuting to school on adolescent weight status. Pediatrics. 2012 Aug;130(2):e296-304. doi: 10.1542/peds.2011-2898. Epub 2012 Jul 16. PMID: 22802608; PMCID: PMC3408684.

REVIEWER: Although the numbers of participants are quite high, there is little information about influencing factors like age, primary/secondary school, distance between school and home, safety of crossings between home and school, having older brothers/sisters to guide younger children etc.

Thank you for your comment. Data on the characteristics of the sample have been included in the results section.

REVIEWER: The introduction includes physical activity information related to household tasks, which of course are important, but not the most appropriate example when assessing PA in children.

Thank you for your comment. More appropriate examples have been written.

“Physical activity is considered as [11] all body movement that spends energy, and therefore, activities such as ride a bike, dance, or active travel are calculated to determine the level of physical activity”

Round 2

Reviewer 3 Report

The authors wrote a manuscript on an important topic. Active transport in youth should be promoted in any country. Despite that I have some comments related to the current version.

Major concern: My most important comment is that I do not see the need to study the study aim. What is the need to know the relationship between transport to school and extracurricular sports? We all know that transport is preferable above motorized transport, but why is it important to know how children go to school and sports; and especially the relation between these two? What is the pain for the society for this matter? What will bring a specific answer?

Minor:

The authors wrote: ‘Commuting to other locations has not been studied in depth. It seems that those students who walked or cycled to school were more likely to use the same type of commuting to other places [20].’ I do not agree with the authors that this study reflect a studying the topic in depth. Are there more valid examples to add information from. Otherwise, change the words ‘in depth’.

The Aibar Solana study is no valid add to this paragraph. I.e. I do not understand the last sentence in this particular paragraph. I suggest to rewrite this part or remove the complete text related to reference 21 out of the manuscript.

Author Response

Dera reviewer, thanks for your remarks, we feel the manuscript has improved a lot with them. Please see our answers to your comments individually below:

Reviewer: Major concern: My most important comment is that I do not see the need to study the study aim. What is the need to know the relationship between transport to school and extracurricular sports? We all know that transport is preferable above motorized transport, but why is it important to know how children go to school and sports; and especially the relation between these two? What is the pain for the society for this matter? What will bring a specific answer?

Authors: We feel that analyzing the relationship between these two modes of transportation for different behaviors will provide us with some useful information, i.e. the pattern of going to school vs the pattern of going to one's chosen extracurricular sports activity  Active commuting may change for a compulsory behavior(such as going to school)  versus a chosen one (going to one's chosen extracurricular sports activity).  Moreover, extracurricular sports activity include exercise, and we could assume that the adolescents may be more incline to active commute or maybe not. We feel active transportation needs to be encouraged and all possibilities should be explored, in particular those who are structured in one's week timetable (school and extracurricular sports activity).

We have modified the introduction slightly to include this rationale.

Minor:

Reviewer: The authors wrote: ‘Commuting to other locations has not been studied in depth. It seems that those students who walked or cycled to school were more likely to use the same type of commuting to other places [20].’ I do not agree with the authors that this study reflect a studying the topic in depth. Are there more valid examples to add information from. Otherwise, change the words ‘in depth’.

The Aibar Solana study is no valid add to this paragraph. I.e. I do not understand the last sentence in this particular paragraph. I suggest to rewrite this part or remove the complete text related to reference 21 out of the manuscript

Authors:. we have removed it completely